# Discordant Genome Assemblies Drastically Alter the Interpretation of Single-Cell RNA Sequencing Data Which Can Be Mitigated by a Novel Integration Method

**DOI:** 10.3390/cells11040608

**Published:** 2022-02-10

**Authors:** Helen G. Potts, Madeleine E. Lemieux, Edward S. Rice, Wesley Warren, Robin P. Choudhury, Mathilda T. M. Mommersteeg

**Affiliations:** 1Burdon Sanderson Cardiac Science Centre, Department of Physiology, Anatomy & Genetics, University of Oxford, Oxford OX1 3PT, UK; helen.potts@new.ox.ac.uk; 2Bioinfo, Plantagenet, ON K0B 1L0, Canada; mlemieux@bioinfo.ca; 3Department of Animal Sciences, Bond Life Sciences Center, University of Missouri, Columbia, MO 65201, USA; edsrice@gmail.com (E.S.R.); warrenwc@missouri.edu (W.W.); 4Division of Cardiovascular Medicine, University of Oxford, Oxford OX3 9DU, UK; robin.choudhury@cardiov.ox.ac.uk

**Keywords:** genome assembly, *Astyanax mexicanus*, integration, seurat, read alignment, non-model organisms, scRNAseq

## Abstract

Advances in sequencing and assembly technology have led to the creation of genome assemblies for a wide variety of non-model organisms. The rapid production and proliferation of updated, novel assembly versions can create vexing problems for researchers when multiple-genome assembly versions are available at once, requiring researchers to work with more than one reference genome. Multiple-genome assemblies are especially problematic for researchers studying the genetic makeup of individual cells, as single-cell RNA sequencing (scRNAseq) requires sequenced reads to be mapped and aligned to a single reference genome. Using the *Astyanax mexicanus*, this study highlights how the interpretation of a single-cell dataset from the same sample changes when aligned to its two different available genome assemblies. We found that the number of cells and expressed genes detected were drastically different when aligning to the different assemblies. When the genome assemblies were used in isolation with their respective annotations, cell-type identification was confounded, as some classic cell-type markers were assembly-specific, whilst other genes showed differential patterns of expression between the two assemblies. To overcome the problems posed by multiple-genome assemblies, we propose that researchers align to each available assembly and then integrate the resultant datasets to produce a final dataset in which all genome alignments can be used simultaneously. We found that this approach increased the accuracy of cell-type identification and maximised the amount of data that could be extracted from our single-cell sample by capturing all possible cells and transcripts. As scRNAseq becomes more widely available, it is imperative that the single-cell community is aware of how genome assembly alignment can alter single-cell data and their interpretation, especially when reviewing studies on non-model organisms.

## 1. Introduction

The use of single-cell RNA sequencing (scRNAseq) technology has greatly increased since it was first developed in 2009 [1]. scRNAseq provides transcriptome information about individual cells, enabling researchers to answer a wide variety of biological questions about topics such as cell–cell heterogeneity, tissue composition and cell-specific gene expression responses to disease and/or injury [2]. Commercialised scRNAseq kits have helped to lower the costs of single-cell experiments, making these experiments more readily accessible to researchers. Indeed, scRNAseq is becoming a routine investigatory approach and has been applied to a range of model and non-model organisms, from the frequently used mouse (*Mus musculus*) [3] to the less commonly used earthworm (*Eisenia andrei*) [4].

The application of scRNAseq to an organism of choice is dependent on the quality and availability of the organism’s reference genome and associated gene annotations. Organisms that are best-suited to scRNAseq have a high-quality reference genome that is well-annotated in the untranslated regions (UTR), as many scRNAseq approaches, such as CEL-seq, MARS-seq, Drop-seq and Chromium, capture mRNA molecules for sequencing via their 3′ polyadenylated tails [5]. Creating high-quality reference genomes was traditionally an expensive, laborious and time-costly undertaking that was reserved exclusively for a few well-funded, large and international consortia [6,7,8,9]. However, developments in high-throughput DNA sequencing, de novo genome assembly technologies and automated genome annotation have significantly decreased the time and cost required for genome assembly construction [10,11]. This has made it feasible for individual labs to sequence and construct a genome assembly, enabling scRNAseq to be applied to any non-model organism of choice [4]. Although the accelerated production and publication of genome assemblies for non-model organisms is very beneficial for a wide range of biomedical research, it can result in multiple-genome assemblies for a given non-model organism to be available concurrently. This is especially problematic for single-cell researchers, as scRNAseq analysis requires sequenced reads to be mapped and aligned to a single reference genome. To date, how interpretation of a single-cell dataset might change depending on the reference to which it is aligned has not been explored.

*A. mexicanus* is a teleost with closely related surface-dwelling and cave-dwelling populations [12]. These separate fish populations arose 10,000-1 million years ago when changes in river levels isolated a series of caves in northeastern Mexico from the surrounding rivers [13,14,15,16]. From this point, the surface- and cave-dwelling populations began to diverge in their evolution as they adapted to their local environment. For example, cave-dwelling populations lost their eyes and pigment [17,18,19,20] and instead gained an altered metabolism that enables them to cope with long periods of fasting due to the scarcity of food available in the caves [21]. Currently, there are two *A. mexicanus* genome assemblies available on Ensembl: v1.0.2, a scaffold-level short-read assembly of an individual from the Pachón cave-dwelling population (listed as Pachón cavefish) [22], and v2.0, a chromosome-level long-read assembly of an F1 hybrid of the Rio Sabinas and Rio Valles surface populations (listed as “Mexican tetra”) [23]. In theory, the v2.0 assembly should produce better results as it is more complete, contiguous, and continuous. To compare the results of using the two different assemblies for alignment, we therefore aligned an *A. mexicanus* single-cell dataset to the v1.0.2 and v2.0 assemblies. We found that the two generated count matrices were fundamentally different, and some results were inconsistent across the two assemblies. To combat this problem, we applied an integration methodology which improved our cell-type annotation and marker gene identification. We propose that our integrated approach will be a useful tool for all researchers using non-model organisms and offers a unifying solution to the problems created by discordant genome assemblies. 

## 2. Materials and Methods

### 2.1. Animal Husbandry

All experimental procedures were performed in accordance with the UK Animals (Scientific Procedures) Act 1986 and institutional guidelines, and conform to the guidelines from Directive 2010/63/EU of the European Parliament on the protection of animals used for scientific purposes. Adult male and female *A. mexicanus* surface fish (1-year) were maintained in the laboratory on a 14/10 h photo-period at 22–25 °C.

### 2.2. Astyanax Mexicanus Heart Dissection and Digestion

Lab-raised *A. mexicanus* surface morphs (*n* = 3) were culled using an overdose of MS222 at 5 g/mL (Sigma, Dorset, UK, Cat. no. A5040). Hearts were isolated and tissue digestion was based on a protocol previously described by Sander et al. (2013) [24]. Ventricles were collected on ice, minced into small pieces, and digested in an Eppendorf tube with 500 µL of digestion buffer. Digestion Buffer was made fresh every digestion and consisted of: 1X PBS, 30 mM Taurine (Sigma, Dorset, UK, cat. no. T8691), 5.5 mM Glucose (Sigma, Dorset, UK, cat. no. G7528), 10 mM 2,3-Butanedione Monoxime (Sigma, Dorset, UK, cat. no. B0753), 10 mM HEPES (Sigma, Dorset, UK, cat. no. H3393), 12.5 µM CaCl_2_ (Sigma, Dorset, UK, cat. no. C4901), 5 mg/mL Collagenase II (Gibco by Life Technologies, Bleiswijk, Netherlands, cat. no. 17101-015), 5 mg/mL Collagenase IV (Gibco by Life Technologies, Bleiswijk Netherlands, cat. no. 17104-019) and 30 µg/mL DNAse I (Sigma, Dorset, UK, cat. no. 10104159001). The CaCl2 concentration was kept at 12.5 µM to ensure adequate cell survival of all cell types present in the heart. Samples were incubated at 32 °C on an Eppendorf thermomixer at 800 rpm. Supernatants were collected every 15 min, neutralised using 1% sheep serum (Sigma, Dorset, UK, cat. no. S3772), and samples were replenished with fresh digestion buffer until all tissue had been digested (approx. 1–2 h). Dissociated cells were filtered through 100 µm filters, spun down (300 g for 5 min at 4 °C), counted, and spun down again before resuspension. Cells were suspended at 2000 cells/µL in DMEM (Life Technologies, Bleiswijk Netherlands, cat. no. 22320-022) plus 10% fetal bovine serum (ThermoFisher, Bleiswijk Netherlands, cat.no. A3840001) before loading onto 10x Chromium Chip B (10x Genomics, Leiden, Netherlands, cat. no. 1000073).

### 2.3. 3′ UTR Extension

The v1.0.2 and v2.0 *A. mexicanus* genome assemblies are poorly annotated in the 3′ UTR (Appendix A). To enable the maximum capture of transcripts during data exploration pending the availability of a better genome annotation, a terminal exon extension algorithm was applied to extend the 3′ UTR annotation. The extension algorithm used a full-length poly(A) RNA-seq sample as a reference and applied the following heuristic:Identify transcripts without 3′ UTR annotation in the Ensembl GTF fileCompare fragment coverage over 100 bp flanking the terminal exonIf median 3′ coverage > median 5′ coverage, extend last exon 100 bp in the 3′ direction and repeat steps 2 & 3 until no further extension occurs.

The extension algorithm was used to create two custom extended GTF files that were used for read counting with the corresponding v1.0.2 and v2.0 genome assemblies (available in Appendix A). In all, 5077/25,489 gene-level annotations on 2530 contigs were extended for v1.0.2, and 8721/27,420 on 25 chromosomes and 1363 contigs were extended for v2.0.

### 2.4. 10x Single-Cell RNA-Sequencing and Analysis

Single-Cell RNAseq libraries were generated using the 10x Chromium Next GEM Single-Cell 3′ v3.1 kit (10x Genomics, Leiden, Netherlands, cat. no. 1000092) and sequenced using the Illumina NextSeq^®^ 500/550 High Output Kit v2 (Illumina, San Diego, United States, cat.no. FC-404-2005). After sequencing, FASTQ files were generated using Cell Ranger mkfastq (v3.0.2). The raw reads were mapped to *Astyanax mexicanus* v1.0.2 and v2.0 genome assemblies using Cell Ranger count with the corresponding 3′ UTR-extended annotations, and two filtered feature matrices were produced. Downstream analysis was performed using the Seurat R package (v4.0.6) [25]. Initial quality control thresholds removed all cells with <50 captured genes and all genes present in <2 cells. Cell filtering thresholds were set based on the average nFeatures and nCounts present in each genome assembly dataset. 

SCTransform [26] was used to normalise, find variable features and scale v1.0.2 and v2.0 datasets individually. The dimensions of the datasets were reduced using Principal Component analysis (PCA) and uniform manifold approximation projection (UMAP). Cells were assigned to clusters using the FindNeighbours and FindClusters functions and the appropriate resolution was chosen using the Clustree package (version 0.4.3) [27]. Marker genes for each cluster were found using. Cell-type annotations were based on marker genes and canonical markers present in v1.0.2 and v2.0 annotations. The v1.0.2 and v2.0 datasets were integrated using 3000 integration features in the Seurat SCTIntegration pipeline. Differential expression analysis was performed using the LR test in the FindMarkers function and visualised using EnhancedVolcano (v1.8.0). Gene Set Enrichment Analysis was carried out by converting *A. mexicanus* genes to their correspondent mouse homologs via BiomaRt (v2.46.3). Any genes which did not have a mouse homolog or mapped to multiple mouse genes were removed, and the final mouse gene lists was tested using the fgsea package (v1.16.0). Conserved genes in the integrated dataset were found using FindConservedMarkers.

## 3. Results

### 3.1. Datasets Generated from the Same Sample Change in Their Fundamental Structure Depending on Genome Assembly and Annotation

To investigate the influence of genome assembly on a single-cell dataset, cardiac ventricular cells from three *A. mexicanus* surface fish were pooled and sequenced. The sequenced reads were aligned to both available genome assemblies with their corresponding gene annotations using Cell Ranger to generate two filtered feature matrices (gene x cell): a v1.0.2 and a v2.0 dataset. To investigate differences in the two datasets, we performed an initial comparison of the Cell Ranger outputs (Table 1). We found that the v1.0.2 and v2.0 datasets had different matrix dimensions, representing differing numbers of cells and expressed genes detected by Cell Ranger. The v2.0 assembly had a >10% higher percentage of sequenced reads that mapped to the transcriptome, resulting in a higher average number of reads and genes detected per cell. The difference in genes present in the matrix had an unexpected impact on cell capture rates; the v2.0 assembly captured an additional 148 cells compared to the v1.0.2. Further comparison between the captured cells showed that both datasets had assembly-specific cells (16 in v1.0.2 and 225 in v2.0). These fundamental differences in cell and feature capture rates required quality control thresholds to be set according to genome-assembly (Figure 1), resulting in a difference of 209 cells available for analysis post-filtering. Therefore, we found that genome assembly choice can alter the fundamental structure of scRNAseq datasets, impacting the number of genes and cells available for downstream analysis.

### 3.2. Dimensional Reduction and Data Visualization Is Robust to Differences in Genome Assembly

To determine whether genome assembly alignment impacted how cells clustered together, the two datasets were separately normalised and scaled using the nonlinear normalisation method SCTransform (Seurat, v3.1.5). The dimensionality of both datasets was reduced using Principal Component Analysis (PCA) and a Uniform Manifold Approximation Projection (UMAP), and cells were clustered together using a graph-based clustering approach. Cells were first embedded in a K-Nearest Neighbour (KNN) graph and then iteratively grouped together with the number of modules optimised using the Louvain algorithm. We found that the resultant datasets are very similar in their structure and number of clusters identified (Figure 2), and thus, genome assembly did not materially impact cell clustering and dimensional reduction.

### 3.3. Incomplete Reference Genomes, When Used in Isolation, Create Problems in Cell-Type Identification and Differential Gene Expression Analysis and Have the Potential to Miss Data

We next sought to identify the cells present in both datasets. We used the receiver operating characteristic (ROC) test to produce a list of gene markers for each cluster in each dataset. Based on these gene markers, we were able to identify the expected different cell types present within the heart, including myocardial, endocardial and epicardial cells, fibroblasts and blood circulating cells (see Appendix A for top cell marker genes used during cell annotation). However, the discordant annotations of the v1.0.2 and the v2.0 genome assemblies presented problems during cell-type identification. We found that the results of differential gene expression analysis were very different between the two datasets, producing two diverging top marker lists for each cardiac cell type, and very few genes were identified as top cell-type markers in both datasets (Table 2). Cell-type identification was further confounded as many canonical cell-type markers, such as α-smooth muscle actin (*acta2*: a marker of pericytes, smooth muscle and myofibroblasts [28,29,30]), are only annotated in one genome assembly.

In addition to the discordant annotation between assemblies resulting in assembly-specific genes, we found that even when genes were present in both genome assemblies, they could show different patterns of expression. We found that *arhgap27*, a gene we have previously shown to be linked to cardiac regeneration [31], has different patterns of expression in the two datasets. The v1.0.2 dataset shows *arhgap27* to be expressed in very few leukocyte cells. On the other hand, the v2.0 dataset shows *arhgap27* to have higher expression levels and suggests that it is also expressed in endothelial cells and cardiomyocytes (Figure 3). Therefore, genome assembly alignment can produce datasets that are inconsistent in terms of the genes present in the matrix and the expression pattern of shared genes.

### 3.4. Genome Assembly Alignment Can Distort Data Interpretation from Specific Cell Types Due to the Problems Created by Underlying Differences in Genome Assemblies

To investigate whether the observed discrepancies between the *A. mexicanus* genome assemblies would impact functional data interpretation, we sought to determine whether any cell types showed assembly-specific differences. Despite similarities in the median number of reads captured per cell, we found that in the v2.0 alignment, there is a higher median number of features detected (Table 1). We therefore sought to determine whether this caused any cell-specific differences in feature-detection rates between the two datasets. We found that although both v1.0.2 and v2.0 datasets showed a similar distribution of features and counts across all clusters (Figure 4a,b), when we compared the number of unique genes detected in each cell type between genome assemblies, a number of v2.0 endothelial cells had a greater number of genes/cell than their v1.0.2 counterparts (Figure 4c,d). To investigate how this increase in median genes/cell might impact the interpretation of endothelial cells we performed differential gene expression analysis between v1.0.2 and v2.0 endothelial cells. We found that when we compared v1.0.2 and v2.0 endothelial cells by logistic regression testing, 2512 genes were upregulated in an assembly-specific manner (Figure 4e). Gene Set Enrichment Analysis (GSEA) of these differentially expressed endothelial genes showed that genes downregulated in response to UV radiation were significantly enriched in the v2.0 dataset (Figure 4f) but not in the v1.0.2 dataset, thus showing that genome alignment can distort the functional interpretation of scRNAseq results, as different results can be generated from the same sample. 

### 3.5. Integration of the Two Datasets Improves Cell-Type Annotation and Maximises the Information That Can Be Obtained from a Single-Cell Dataset

To overcome the problems posed by the discordant genome assemblies, we integrated both datasets together using the SCTIntegration pipeline. This produced an integrated dataset in which cells that originated from either the v1.0.2 or the v2.0 were present and treated as unique. We found that SCTIntegration produced a UMAP with all of the expected cell types (Figure 5a) and that cells clustered together regardless of genome alignment (Figure 5b). The integrated dataset included all 241 genome-specific cells (Figure 5c). This increased the size of a wide range of cell-type clusters such as erythrocytes, endothelial cells, cardiomyocytes, leukocytes, smooth muscle and fibroblasts, ensuring the maximal amount of data were captured in the final integrated dataset. 

During cluster annotation, the integrated dataset enabled accurate cell-type identification, as canonical markers that are only annotated in one assembly, such as *acta2*, could be used simultaneously with the integrated dataset cluster marker genes to annotate each cluster (Figure 5d). We found that the integrated dataset resulted in the inclusion of 4311 assembly-specific genes for v1.0.2, and 5638 for v2.0, that can be utilised during cell annotation. Additionally, we found that 525 cells from the v1.0.2 dataset and 765 cells from the v2.0 dataset were annotated differently in the integrated dataset. Specifically, we found that the integrated dataset allowed more accurate annotation of doublets (see Appendix A for transcriptional profile of doublets), as many of the cells that changed annotation were found in doublet clusters in the integrated dataset (25.5% for v1.0.2 cells and 68.5% for v2.0 cells) (Figure 5e). 

We finally used the integrated dataset to calculate a list of conserved gene symbols present in both genome alignments that will act as a robust list of cell-type markers of previously uncharacterised *A. mexicanus* cell types (Appendix A). The integrated dataset provided a solution in which all possible genes and transcripts could be probed simultaneously for optimal cell-type annotation.

## 4. Discussion

In this study, we use single-cell RNA sequencing data and two published reference assemblies from *A. mexicanus* to show, for the first time, that the same set of scRNAseq reads can produce different results when aligned to different genome assemblies, generating differences in matrix dimensions, gene-expression patterns and cell-type identification. Critically, the finding that only v2.0 endothelial cells show significant enrichment for genes responsive to UV radiation confirms our hypothesis that genome assembly choice can impact data interpretation. To overcome the problems posed by multiple-genome assemblies that are discordantly annotated, we propose the alignment of scRNAseq samples to all available assemblies, followed by integration, to create a finalised dataset for use in downstream analysis.

Our proposed methodology will be a useful tool for researchers using non-model organisms with more than one available genome assembly. Within the *A. mexicanus* field, there is no consensus as to which genome assembly should be used for sequencing experiments. Recent publications have used v1.0.2 [31,32], v2.0 [33], an archived version of the genome (astmex1, Ensembl 87 gene model) [34], and even a new Pachón build only available on NCBI [35]. Such inconsistent use of the *A. mexicanus* genome assemblies will hinder research progress by introducing artefacts. Our finding that the pattern of *arhgap27* expression completely changes with genome alignment emphasises that an integrated use of all genome assemblies is essential when characterizing novel cell types and gene expression patterns. Furthermore, our finding that v2.0 endothelial cells were significantly enriched for genes annotated in the UV response hallmark pathway suggests that if assemblies continue to be used in isolation, this could lead to a problem wherein results found using v1.0.2 are not transferable to v2.0 (and vice versa). Therefore, we propose that both v1.0.2 and v2.0 assemblies be used for all *A. mexicanus* genomic studies until a better-curated, accurate and well-annotated reference genome is available. 

This work is relevant not only to *A. mexicanus* researchers but also to the wider scRNAseq community, in particular for groups that work with non-model organisms with genomes that continue to be produced at a rapid pace using a variety of different technologies. It is likely that the observed genome discordancies that arise in the v1.0.2 and v2.0 *A. mexicanus* assemblies are a result of the different sequencing technologies used during genome construction and the post-sequencing assembly and annotation algorithms. The Pachόn cavefish assembly (v1.0.2) was assembled from DNA from the heart, liver, spleen and gill of a single 7-year-old adult female using short reads and mate-paired libraries, with a de Bruijn graph assembler [36], in 2014. The surface fish assembly is a more recent assembly from a single adult female surface fish that was constructed using single-molecule-long reads, optical mapping, and a genetic linkage map. Long read constructions are better able to resolve heterogeneous, highly repetitive regions of the genome, which may explain why v2.0 endothelial cells showed significant enrichment of genes responsive to UV radiation that was absent in v1.0.2 cells. Although it is beyond the scope of individual labs to resolve genome assembly or gene annotation inconsistencies, single-cell researchers should be aware of how the assembly and annotation of their genome might be impacting their data interpretation and capture of specific cell types and genes of interest. Therefore, we recommend that researchers working with multiple incomplete reference genomes align their scRNAseq data to all constructed genome assemblies to ensure that no cells or genes are unnecessarily excluded from the final dataset. We propose that researchers use our methodology until a high-quality reference genome is available for their non-model organism of choice.

Finally, our applicable integration methodology not only provides a unifying approach to the problems posed by multiple-genome assemblies, but it also ensures that the maximum amount of information can be extracted from a single-cell sample. Integrating the v1.0.2 and v2.0 datasets allowed the expression of all assembly-specific genes to be explored during analysis of the final dataset. This led to the inclusion of more than 9k assembly-specific genes that could be used during cluster annotation, thereby ensuring that cell types can be attributed as accurately as possible. Additionally, the integrated dataset includes 241 cells that would have been lost if the *A. mexicanus* genome assemblies were used in isolation. This approach would therefore be very beneficial to researchers trying to detect rare cell types, as maximising the number of cells that are retained for analysis decreases the chance that any important small cell populations are lost from the final dataset. 

We do suggest that our integrated approach should mainly be used for cell-type annotation and marker-gene identification. Once the integrated dataset has been created and cell types have been accurately identified, researchers should subset the integrated dataset into its component genome-assembly datasets to perform downstream analysis. Additionally, we used a 3′ UTR extension algorithm to create a custom 3′ UTR extended gtf file for read counting, to try to offset the incomplete 3′ UTR annotation of the *A. mexicanus* genome assemblies. This maximised our ability to capture transcripts and identify possible leads from our available data. However, genes of interest identified using our 3′ UTR extension algorithm should be treated with caution, and would require confirmation of gene expression levels using additional methods such as qPCR and in situ hybridization.

In conclusion, we propose a novel solution to address the vexing problems posed by multiple-genome assemblies that are discordantly annotated. Our methodology opens the door to applying scRNAseq to non-model organisms, even those with multiple, fragmentary genome assemblies. This could change how we approach answering biological questions as, all too often, model organisms that are not ideally suited to dissecting a particular disease or biological process are used for pragmatic reasons, as they possess polished and well-annotated reference genomes. We hope that our approach will help researchers to design their scRNAseq using their organism of choice based on how well it recapitulates the question at hand, rather than being limited by incomplete genome assemblies.

## Figures and Tables

**Figure 1 cells-11-00608-f001:**
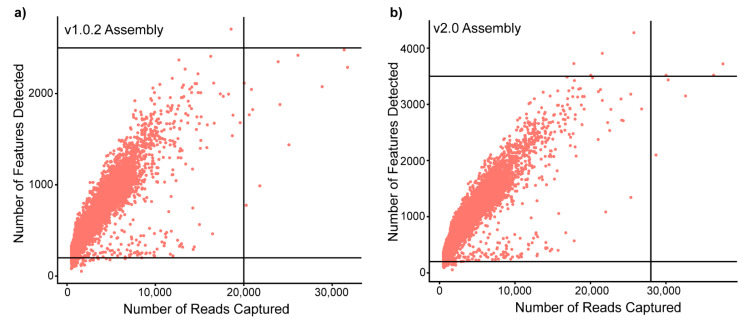
**Genome assembly can alter the nFeatures and nCounts of a scRNaseq dataset, impacting the number of cells and genes available for downstream analysis.** Scatter plots of the number of counts (reads) vs. number of features (genes) in each captured cell when aligned to: (**a**) the v1.0.2 genome assembly and; (**b**) the v2.0 assembly. The lines on each scatter plot represent the quality control thresholds that were set for each genome assembly, highlighting the increase in the number of features and reads captured in the v2.0 dataset vs. v1.0.2 dataset.

**Figure 2 cells-11-00608-f002:**
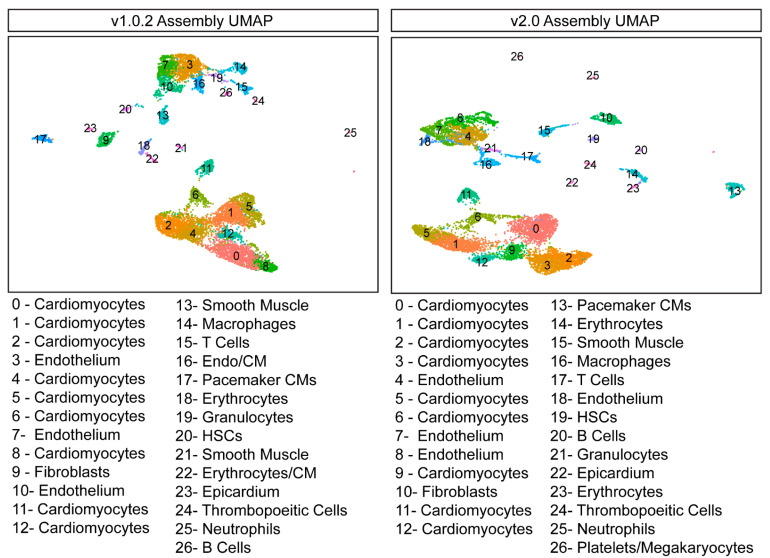
**Genome assembly does not impact dimension reduction and cell clustering.** UMAPs generated from the v1.0.2 and v2.0 datasets are very similar. The v1.0.2 B cell cluster was manually annotated. HSCs—Hematopoietic Stem Cells.

**Figure 3 cells-11-00608-f003:**
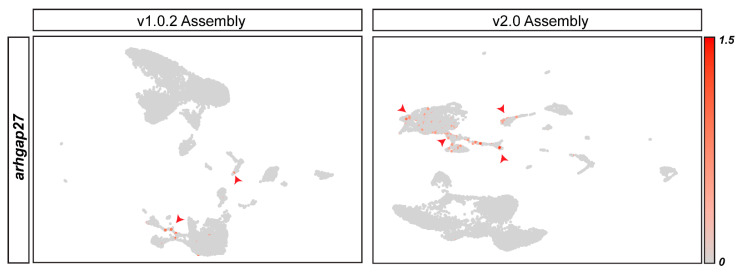
***arhgap27*****is discordantly annotated in genome assembly v1.0.2 and v2.0.** FeaturePlot shows the distribution of *arhgap27^+^* cells on the v1.0.2 and v2.0 datasets, as highlighted in red.

**Figure 4 cells-11-00608-f004:**
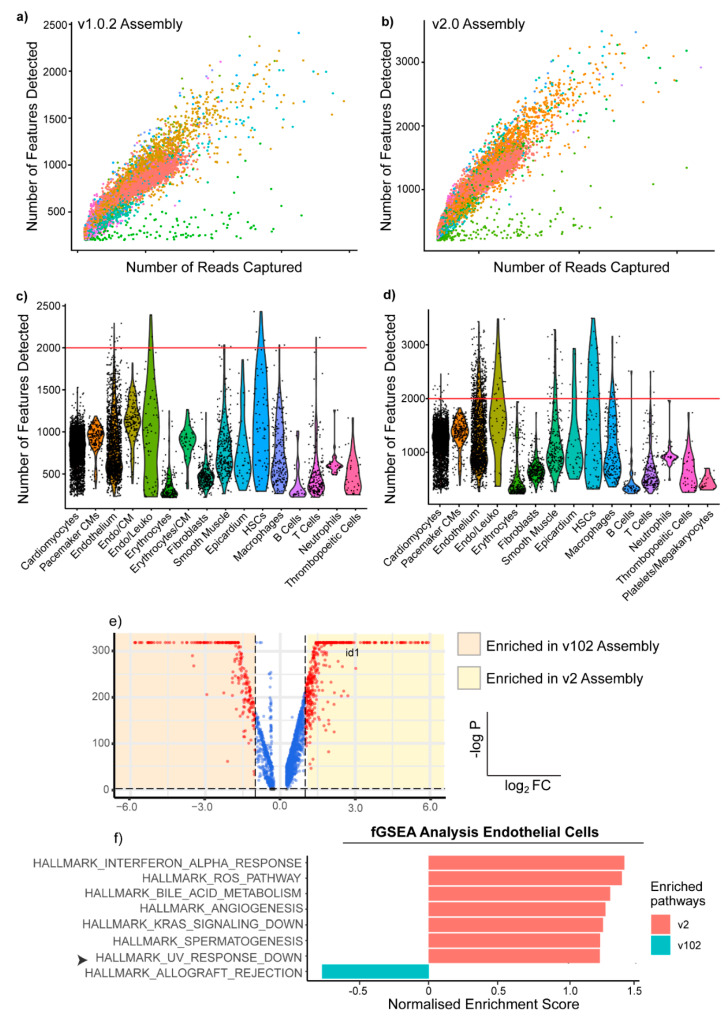
**Genome alignment choice produces assembly-specific results due to discordant genome annotations.** Scatter plots showing the distribution of counts and features amongst the identified cell clusters in (**a**) the v1.0.2 genome assembly and (**b**) the v2.0 assembly. Violin plots showing the distribution of the number of genes detected/cell in each major cardiac cell type in (**c**) the v1.0.2 assembly and (**d**) the v2.0 assembly. The red line is drawn at 2000 features/cell to show how many v2.0 endothelial cells express a greater number of unique genes. (**e**) Volcano Plot showing the results of differential expression analysis of v.1.0.2 vs. v2.0 endothelial cells. *id1* is upregulated in the v2.0 endothelial cells and is annotated in the Hallmark pathway of genes downregulated in response to UV radiation. (**f**) GSEA analysis showing the top hallmark terms per dataset, with the pathway that reached the 0.25 false discovery rate (FDR) threshold highlighted by the black arrow, showing that genes annotated in the hallmark pathway as downregulated in response to UV radiation are significantly enriched in v2.0 endothelial cells.

**Figure 5 cells-11-00608-f005:**
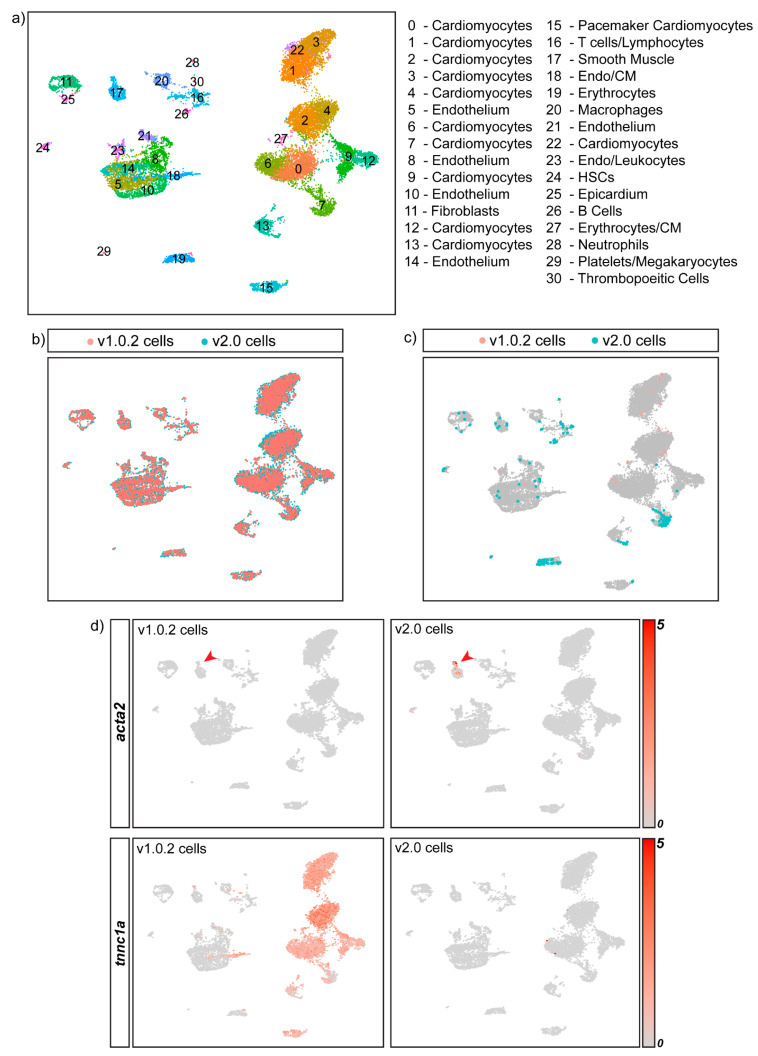
**Integrating the v1.0.2 and v2.0 datasets improves cell-type identification and maximises the number of cells included in the final dataset.** (**a**) Annotated UMAP of the integrated dataset showing all of the expected cell types of the heart. CM- Cardiomyocytes (**b**) Cells coloured according to genome assembly on the UMAP shows that SCTIntegration produces a UMAP in which cells from both assemblies cluster together. (**c**) 241 cells present in the integrated dataset are genome-assembly-specific. These cells fall within a range of cell-type clusters such as erthyrocytes, endothelial cells, cardiomyocytes and fibroblasts, and would be excluded if either the v1.0.2 or the v2.0 assembly was used in isolation. (**d**) FeaturePlot split by genome assembly was used to plot 2 separate UMAPs in which each UMAP contains cells from either the v2.0 dataset or the v1.0.2 dataset, revealing the cluster expression of assembly-specific marker genes such as *acta2* and *tnnc1a* in the integrated dataset. (**e**) 525 cells from the v1.0.2 dataset and 765 cells from the v2.0 dataset were annotated differently in the integrated dataset. Many of these cells were found to cluster in doublet clusters and were re-annotated as doublets, as indicated by the black arrows.

**Table 1 cells-11-00608-t001:** Table of differences in the 2 datasets generated using v1.0.2 and v2.0 genome assemblies.

Assembly Readouts	v1.0.2 Assembly	v2.0 Assembly
Matrix Dimensions	25,489 genes, 8870 cells	27,420 genes, 9018 cells
Reads Mapped to Genome	72.6%	74.1%
Reads Mapped to Transcriptome	41.7%	52.4%
Median Reads/Cell	30,901	30,394
Median Genes/Cell	794	1201
Assembly-specific cells	16	225
Assembly-specific genes	4311	5638
nCount vs. nFeature Correlation	0.84	0.88
Quality Control Thresholds	nFeatures: 200–2500	nFeatures: 200–3500
	nCounts: <20,000	nCounts <28,000
% Cells Passed Filtering	98.2%	98.9%
Cell Numbers Post-Filtering	8717 cells	8926 cells
% Genes Passed Filtering	64.3%	63.9%
Gene Numbers Post-Filtering	16,408 genes	17,528 genes
PCA Dimensions	30	30
Clustering Resolution	1.25	1.25
Number of Clusters Found	26	27

**Table 2 cells-11-00608-t002:** **Table of the top 10 marker genes identified using FindMarkers for the major cell types present in the heart.** Gene symbols have been provided where possible. For genes where gene symbol annotation is not available or multiple Ensembl IDs have the same gene symbol, Ensembl IDs have been provided. Genes identified in both genome assemblies are highlighted in red. * denotes genes that are assembly-specific.

Cell Type	v1.0.2 Assembly	v2.0 Assembly
Cardiomyocytes	*ENSAMXG00005008576 **	*myh7l **
	*actc1a*	*ENSAMXG00000004797 **
	* tnnc1a *	*nme2b.1 **
	*cox6a2 **	* TNNC1 *
	* aldoaa *	* aldoab *
	*zgc:193541*	* idh2 *
	*ENSAMXG00005013223 **	*cox7b*
	* IDH2 *	*slc25a5*
	*atp5mc3a*	*cox7c*
	*atp5if1a **	*tnnt2b*
Endothelium	*ENSAMXG00005007750 **	*lyve1a*
	*ENSAMXG00005016906 **	*ENSAMXG00000041928 **
	*ENSAMXG00005003412 **	*krt18a.1*
	*ENSAMXG00005021204 **	*rgs5b*
	*plpp2a*	*ENSAMXG00000036379 **
	*ENSAMXG00005022026 **	*il13ra2*
	*krt5*	*sat1a.2*
	*Ucmaa **	*serpinh1b*
	*ENSAMXG00005012084 **	*her6*
	*ENSAMXG00005004741 **	*ENSAMXG00000035697 **
Fibroblasts	*thbs1a*	*ccl25b*
	* tcf21 *	*rbp4*
	*lxn*	*apoeb*
	*kcne4*	*pmp22a*
	*mustn1a*	*tagln*
	*ENSAMXG00005006660 **	* TCF21 *
	*pltp*	*dcn*
	*clec19a*	*col1a2*
	*BAMBI **	*anxa1a (ENSAMXG00000035597)*
	*hmx4*	*sostdc1a*
Neutrophils	*ENSAMXG00005022612 **	*ENSAMXG00000006427 **
	* c6ast1 *	* mmp9(ENSAMXG00000007722) *
	*ENSAMXG00005012967 **	*ENSAMXG00000035474 **
	*ENSAMXG00005007030 **	*LECT2*
	*ENSAMXG00005022013 **	* c6ast1 *
	*adam8a*	*ENSAMXG00000037167 **
	*ltb4r*	*cebp1*
	*ENSAMXG00005015365 **	*ENSAMXG00000001798 **
	* mmp9 *	*Scinlb*
	*ENSAMXG00005024801 **	*ENSAMXG00000034260 **
T cells	* pfn1 *	*ENSAMXG00000036068 **
	* laptm5 *	* pfn1 *
	* cxcr4b *	* laptm5 *
	*ENSAMXG00005014236 **	*ctsl.1 (ENSAMXG00000029871)*
	*coro1a*	* cxcr4b *
	*ENSAMXG00005012967 **	*rac2 **
	* rgs13 *	*dusp2*
	*PTPRC*	* rgs13 *
	*runx3*	*cotl1*
	*ENSAMXG00005022013 **	*ENSAMXG00000001798 **
B cells	*ENSAMXG00005001652 **	*ENSAMXG00000033936 **
	*ENSAMXG00005007434 **	*zgc:194275*
	* cd37 *	*ENSAMXG00000029163 **
	*ENSAMXG00005014280 **	*ENSAMXG00000038512 **
	*si:dkey-24p1.1*	*ENSAMXG00000006777 **
	*ENSAMXG00005006484 **	* cd37 *
	*ENSAMXG00005014291 **	*ENSAMXG00000036191 **
	*ENSAMXG00005012813 **	*ENSAMXG00000034153 **
	*ENSAMXG00005000610 **	*ENSAMXG00000043949 **
	*ENSAMXG00005002435 **	*ENSAMXG00000043088 **
Erythrocytes	* hbaa2 *	*ENSAMXG00000029151 **
	*ENSAMXG00005017042 **	* hbaa2 (ENSAMXG00000029181) *
	* wu:fj16a03 *	*ENSAMXG00000037273 **
	* cahz *	*hbba2*
	*nt5c2l1*	*HBE1 (ENSAMXG00000037475) **
	*ENSAMXG00005020328 **	*si:ch211-250g4.3 **
	*ENSAMXG00005017060 **	*si:ch211-103n10.5*
	*mt2.2*	* Cahz *
	*zgc:163057 **	*slc4a1a*
	*ENSAMXG00005017061 **	* wu:fj16a03 *
Epicardium	*ENSAMXG00005022849 **	*ENSAMXG00000036050 **
	* tcf21 *	* TCF21 *
	*ENSAMXG00005007716 **	*Cfd*
	*ENSAMXG00005012482 **	*c3a.1 **
	*zgc:158846*	*ENSAMXG00000036137 **
	*ENSAMXG00005022791 **	*igfbp5a*
	*ENSAMXG00005009039 **	*stmn1a*
	*ENSAMXG00005008245 **	* wt1b *
	*ENSAMXG00005022313 **	*glis2a*
	* wt1b *	*slc29a1a*
Smooth Muscle	* si:dkey-57k2.6 *	*CASP6*
	*ENSAMXG00005003735 **	*Tagln*
	* thbs1b *	*angptl7*
	*ENSAMXG00005011018 **	*ENSAMXG00000031755 **
	*TPM1*	*acta2 **
	* anxa1a *	* si:dkey-57k2.6 *
	*ITIH3*	*rbp4*
	*ENSAMXG00005005681 **	*sox9b*
	*ENSAMXG00005018089 **	*anxa1a (ENSAMXG00000035597)*
	*thbs4a*	*thbs1b*
Dendritic Cells/Macrophages	*ENSAMXG00005011614 **	*ENSAMXG00000036068 **
	*ENSAMXG00005002001 **	*ccl35.1 **
	*ENSAMXG00005021693 **	*ENSAMXG00000037572 **
	*ccl34a.3 **	* cd74a *
	* cxcr4b *	*ENSAMXG00000004394 **
	*ENSAMXG00005014236 **	*cxcl8a*
	* cd74a *	* cxcr4b *
	*ENSAMXG00005001734 **	*ENSAMXG00000042210 **
	*ENSAMXG00005009773 **	*si:dkey-5n18.1*
	*c1qb*	*il1b (ENSAMXG00000035729) **

## Data Availability

The single-cell RNAseq data files have been deposited in GEO under accession number GSE194093. The code used to generate the plots in the paper can be found in github (https://github.com/helenpotts/Astyanax_mexicanus_genome_assembly_analysis, accessed on 22 January 2022).

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
