# Peer review of "Discordant Genome Assemblies Drastically Alter the Interpretation of Single-Cell RNA Sequencing Data Which Can Be Mitigated by a Novel Integration Method"

_cells, 2022, doi:10.3390/cells11040608_

Round 1

Reviewer 1 Report

This manuscript showed that novel method for integration of scRNA-seq data focusing on genome assembly.
The bias of genome assembly is important concern. This concern would be a problem that cannot be ignored by researchers performing scRNA-seq.
However, there are several concerns for the publication.

Comments
1. The authors should perform the comparative analysis using the another public scRNA-seq dataset to show the availability of the novel method.

2. The authors showed the differences of cell annotation in the clustering between the two assemble method. In fact, the reviewer wonder how many singe-cells were altered by novel method. The analysis lacked the quantitative and statistical viewpoints. When each single-cell was linked the clustering information of the two methods, how many cells have changed their cell type annotation?

3. The authors described that they will deposit the dataset in the GEO database and make the code available on github. However, it should be deposited, and included in this manuscript immediately. If publicity and reproducibility cannot be guaranteed, it will be a problem for the publication.

4. In materials and methods section, 2.3 and 2.4. probably have a lot of periods.

Author Response

We thank the reviewer for reviewing our manuscript in detail and for their useful comments. We have addressed their points in the manuscript and below.

1) The authors should perform the comparative analysis using the another public scRNA-seq dataset to show the availability of the novel method.

We thank the reviewer for their comments and understanding of the applicability of our method. Unfortunately, scRNAseq is still a relatively new technique, especially in its application to non-model organisms and, as such, there are not yet any publicly available comparable scRNAseq datasets. Up to this point, as we have seen in the A. mexicanus field, when faced with multiple genome assemblies researchers have historically picked one assembly to align their data to and ignored other available assemblies. We hope that as the scRNAseq community begins to understand the influence of genome assembly on scRNAseq datasets based on our work, this will change and in the future, scRNAseq datasets aligned to multiple assemblies will appear. For now, we cannot apply our method to another species, although we do agree that this will be a worthwhile pursuit when such datasets are available.

2) The authors showed the differences of cell annotation in the clustering between the two assemble method. In fact, the reviewer wonder how many single-cells were altered by novel method. The analysis lacked the quantitative and statistical viewpoints. When each single-cell was linked the clustering information of the two methods, how many cells have changed their cell type annotation?

Reviewer 1 makes an excellent point and so for the revisions we have investigated how many cells changed cluster from genome assembly annotation to the integrated dataset. We found that many cells changed cluster which we have included in Results section 3.5. Additionally, we found that doublets were more easily identified in the integrated dataset. We have added a new panel (E) to Figure 5 to show the new cluster locations for the cells that have changed annotation and to highlight the increase in doublet cluster size. We have also added a heatmap to the Supplementary Material (Supplementary Figure 3) to show that the transcriptional profile of the identified doublets mirror the profiles of more than one cardiac cell type. To the text e have added:

Lines 302-309:

“Additionally, we found that 525 cells from the v1.0.2 dataset and 765 cells from the v2.0 dataset were annotated differently in the integrated dataset. Specifically, we found that the integrated dataset allowed more accurate annotation of doublets (see Supplementary Figure 3 for transcriptional profile of doublets) as many of the cells that changed annotation were found in doublet clusters in the integrated dataset (25.5% for v1.0.2 cells and 68.5% for v2.0 cells) (Figure 5e).”

Lines 293-296 (Figure Legend):

“(E) 525 cells from the v1.0.2 dataset and 765 cells from the v2.0 dataset were annotated differently in the integrated dataset. Many of these cells were found to cluster in doublet clusters and were re-annotated as doublets, as indicated by the black arrows.”

3) The authors described that they will deposit the dataset in the GEO database and make the code available on github. However, it should be deposited, and included in this manuscript immediately. If publicity and reproducibility cannot be guaranteed, it will be a problem for the publication.

We had planned to deposit the data during manuscript revisions. This has now been completed and its GEO accession number is GSE194093. As is standard practice, the GEO submission is currently private and will be made public when the manuscript is accepted for publication. Reviewers can access the GEO submission using ozspmqemjfeljml

We have updated the data availability section with the following (Lines 405-408):

“The single cell RNAseq data files have been deposited in GEO under accession number GSE194093. The code used to generate the plots in the paper can be found in github (https://github.com/helenpotts/Astyanax_mexicanus_genome_assembly_analysis).”

4) In materials and methods section, 2.3 and 2.4. probably have a lot of periods.

We have removed the unnecessary full stops from the 2.3 and 2.4 subsections.

Reviewer 2 Report

The authors proposed a novel solution to address the vexing problems posed by multiple genome assemblies that are discordantly annotated. Their methodology opens the door to applying scRNAseq to non-model organisms, even those with multiple, fragmentary genome assemblies. There are several major problems:

  1. The authors said “The single cell RNAseq data files will be deposited in GEO under accession number. The code used to generate the plots in the paper will be found on github.” They should do it before submission.
  2. In Figure 1, why did the authors use different cutoffs in a and b?
  3. How did the authors do the cell type annotation? Many cell clusters were located in very different locations but they were annotated with the same cell type.
  4. In Table 2, the authors should use either IDs or Symbols.
  5. In Figure 5 A, why so many cells were not annotated?
  6. In Figure 5 D, “FeaturePlot split by genome assembly shows that marker genes that are assembly-specific, like acta2 and tnnc1a, can be used for cluster identification in the integrated dataset.” What did the authors mean? The cells were real, not some tricks.

Author Response

We thank the reviewer for reviewing our manuscript in detail and for their useful comments. We have addressed their points in the manuscript and below.

1. The authors said “The single cell RNAseq data files will be deposited in GEO under accession number. The code used to generate the plots in the paper will be found on github.” They should do it before submission.

We had planned to deposit the data during manuscript revisions. This has now been completed and its GEO accession number is GSE194093. As is standard practice, the GEO submission is currently private and will be made public when the manuscript is accepted for publication. Reviewers can access the GEO submission using ozspmqemjfeljml

We have updated the data availability section with the following (Lines 405-408):

“The single cell RNAseq data files have been deposited in GEO under accession number GSE194093. The code used to generate the plots in the paper can be found in github (https://github.com/helenpotts/Astyanax_mexicanus_genome_assembly_analysis).”

2. In Figure 1, why did the authors use different cutoffs in a and b?

The quality control thresholds were set according to the average number of features detected/cell and number of reads captured/cell. Due to differences in underlying genome annotation, the v2.0 assembly had a 10% increase in the number of reads that mapped to the transcriptome, resulting in a higher average for number of features detected and reads captured per cell, meaning that different cut offs needed to be set to exclude the outlier cells in each dataset. We have added the below lines to results section 3.1 to make this clearer to readers:

Lines 167-169:

“The v2.0 assembly had a >10% higher percentage of sequenced reads that mapped to the transcriptome, resulting a higher average number of reads and genes detected per cell.”

3. How did the authors do the cell type annotation? Many cell clusters were located in very different locations but they were annotated with the same cell type.

We thank reviewer 2 for the comments and realised that some of the cell clusters had been mislabelled in Figure 2. We have rectified the errors in Figure 2 and added an additional figure to the Supplementary Material (Supplementary Figure 2), which is a DotPlot, representing the top cell type markers that were used during cell annotation for each genome assembly.

4. In Table 2, the authors should use either IDs or Symbols.

In both v1.0.2 and v2.0 A. mexicanus genome assemblies, gene symbols are not available for all annotated genes and some gene symbols have more than one Ensembl ID . As we felt that gene symbols are more informative markers than their corresponding Ensembl IDs, we have provided gene symbols where possible and provided the Ensembl IDs if gene symbols were not available. We have also added the Ensembl IDs to the gene symbols that have multiple associated Ensembl IDs. We thank reviewer 2 for commenting on this and have amended the figure legend for table 2 to make this clearer to readers.

Lines 218-2207

“Gene symbols have been provided where possible. For genes where gene symbol annotation is not available or multiple Ensembl IDs have the same gene symbol, Ensembl IDs have been provided.”

5. In Figure 5 A, why so many cells were not annotated?

We had used quotation marks in Figure 5A to indicate that the above cell label was repeated. We realise it was not clear and have amended Figure 5A to remove the quotation marks and replaced them with cell labels.

6. In Figure 5 D, “FeaturePlot split by genome assembly shows that marker genes that are assembly-specific, like acta2 and tnnc1a, can be used for cluster identification in the integrated dataset.” What did the authors mean? The cells were real, not some tricks.

The aim of Figure 5D was to make the point that integrating the 2 datasets allowed assembly-specific marker genes to be used during cell annotation. We split the FeaturePlot into 2 UMAPs where UMAP contains cells from either the v1.0.2or the v2.0 dataset. Although the cells were real, due to the poor quality of the A. mexicanus genome assemblies, the absence of a cell type marker in one assembly is unlikely to be real and it is much more likely that the gene is simply not annotated in that assembly rather than not expressed in those cells. We split the UMAPs to show that in the overall integrated dataset, we could therefore use a combination of assembly-specific markers as well as top cluster markers during cell type annotation.

To make this point clearer, the following has been added to the figure legend for Figure 5:

Lines 289-293:

“(D) FeaturePlot split by genome assembly was used to plot 2 separate UMAPs in which each UMAP contains cells from either the v2.0 dataset or the v1.0.2 dataset, revealing the cluster expression of assembly-specific marker genes, like acta2 and tnnc1a, in the integrated dataset.”

Also the following sentence on lines 293-296 was amended:

“During cluster annotation, the integrated dataset enabled accurate cell type identification as canonical markers that are only annotated in one assembly, like acta2, could be used simultaneously with the integrated dataset cluster marker genes to annotate each cluster (Figure 5d).”

Round 2

Reviewer 1 Report

The authors addressed all my concern.

Reviewer 2 Report

The authors have answered my questions.